# Semi-supervised Learning with GANs: Manifold Invariance with Improved Inference

**Abhishek Kumar**[*]
IBM Research AI
Yorktown Heights, NY
abhishk@us.ibm.com

**Prasanna Sattigeri**[*]
IBM Research AI
Yorktown Heights, NY
psattig@us.ibm.com

**P. Thomas Fletcher**
University of Utah
Salt Lake City, UT
fletcher@sci.utah.edu

## Abstract

Semi-supervised learning methods using Generative adversarial networks (GANs) have shown promising empirical success recently. Most of these methods use a shared discriminator/classifier which discriminates real examples from fake while also predicting the class label. Motivated by the ability of the GANs generator to capture the data manifold well, we propose to estimate the tangent space to the data manifold using GANs and employ it to inject invariances into the classifier. In the process, we propose enhancements over existing methods for learning the inverse mapping (i.e., the encoder) which greatly improves in terms of semantic similarity of the reconstructed sample with the input sample. We observe considerable empirical gains in semi-supervised learning over baselines, particularly in the cases when the number of labeled examples is low. We also provide insights into how fake examples influence the semi-supervised learning procedure.

## 1 Introduction

Deep generative models (both implicit [11, 23] as well as prescribed [16]) have become widely popular for generative modeling of data. Generative adversarial networks (GANs) [11] in particular have shown remarkable success in generating very realistic images in several cases [30, 4]. The generator in a GAN can be seen as learning a nonlinear parametric mapping $g : Z \to X$ to the data manifold. In most applications of interest (e.g., modeling images), we have $\dim(Z) \ll \dim(X)$. A distribution $p_z$ over the space $Z$ (e.g., uniform), combined with this mapping, induces a distribution $p_g$ over the space $X$ and a sample from this distribution can be obtained by ancestral sampling, i.e., $z \sim p_z, x = g(z)$. GANs use adversarial training where the discriminator approximates (lower bounds) a divergence measure (e.g., an $f$-divergence) between $p_g$ and the real data distribution $p_x$ by solving an optimization problem, and the generator tries to minimize this [28, 11]. It can also be seen from another perspective where the discriminator tries to tell apart real examples $x \sim p_x$ from *fake* examples $x_g \sim p_g$ by minimizing an appropriate loss function[10, Ch. 14.2.4] [21], and the generator tries to generate samples that maximize that loss [39, 11].

One of the primary motivations for studying deep generative models is for semi-supervised learning. Indeed, several recent works have shown promising empirical results on semi-supervised learning with both implicit as well as prescribed generative models [17, 32, 34, 9, 20, 29, 35]. Most state-of-the-art semi-supervised learning methods using GANs [34, 9, 29] use the discriminator of the GAN as the classifier which now outputs $k + 1$ probabilities ($k$ probabilities for the $k$ *real* classes and one probability for the *fake* class).

When the generator of a trained GAN produces very realistic images, it can be argued to capture the data manifold well whose properties can be used for semi-supervised learning. In particular, the

---

[*]Contributed equally.

tangent spaces of the manifold can inform us about the desirable invariances one may wish to inject in a classifier [36, 33]. In this work we make following contributions:

- We propose to use the tangents from the generator's mapping to automatically infer the desired invariances and further improve on semi-supervised learning. This can be contrasted with methods that assume the knowledge of these invariances (e.g., rotation, translation, horizontal flipping, etc.) [36, 18, 25, 31].
- Estimating tangents for a real sample $x$ requires us to learn an encoder $h$ that maps from data to latent space (inference), i.e., $h : X \to Z$. We propose enhancements over existing methods for learning the encoder [8, 9] which improve the semantic match between $x$ and $g(h(x))$ and counter the problem of *class-switching*.
- Further, we provide insights into the workings of GAN based semi-supervised learning methods [34] on how fake examples affect the learning.

## 2 Semi-supervised learning using GANs

Most of the existing methods for semi-supervised learning using GANs modify the regular GAN discriminator to have $k$ outputs corresponding to $k$ real classes [38], and in some cases a $(k+1)$'th output that corresponds to *fake* samples from the generator [34, 29, 9]. The generator is mainly used as a source of additional data (*fake* samples) which the discriminator tries to classify under the $(k+1)$th label. We propose to use the generator to obtain the tangents to the image manifold and use these to inject invariances into the classifier [36].

### 2.1 Estimating the tangent space of data manifold

Earlier work has used contractive autoencoders (CAE) to estimate the local tangent space at each point [33]. CAEs optimize the regular autoencoder loss (reconstruction error) augmented with an additional $\ell_2$-norm penalty on the Jacobian of the encoder mapping. Rifai et al. [33] intuitively reason that the encoder of the CAE trained in this fashion is sensitive only to the tangent directions and use the dominant singular vectors of the Jacobian of the encoder as the tangents. This, however, involves extra computational overhead of doing an SVD for every training sample which we will avoid in our GAN based approach. GANs have also been established to generate better quality samples than prescribed models (e.g., reconstruction loss based approaches) like VAEs [16] and hence can be argued to learn a more accurate parameterization of the image manifold.

The trained generator of the GAN serves as a parametric mapping from a low dimensional space $Z$ to a manifold $\mathcal{M}$ embedded in the higher dimensional space $X$, $g : Z \to X$, where $Z$ is an open subset in $\mathbb{R}^d$ and $X$ is an open subset in $\mathbb{R}^D$ under the standard topologies on $\mathbb{R}^d$ and $\mathbb{R}^D$, respectively $(d \ll D)$. This map is not surjective and the range of $g$ is restricted to $\mathcal{M}$.[2] We assume $g$ is a smooth, injective mapping, so that $\mathcal{M}$ is an embedded manifold. The Jacobian of a function $f : \mathbb{R}^d \to \mathbb{R}^D$ at $z \in \mathbb{R}^d$, $J_z f$, is the matrix of partial derivatives (of shape $D \times d$). The Jacobian of $g$ at $z \in Z$, $J_z g$, provides a mapping from the tangent space at $z \in Z$ *into* the tangent space at $x = g(z) \in X$, i.e., $J_z g : T_z Z \to T_x X$. It should be noted that $T_z Z$ is isomorphic to $\mathbb{R}^d$ and $T_x X$ is isomorphic to $\mathbb{R}^D$. However, this mapping is not surjective and the range of $J_z g$ is restricted to the tangent space of the manifold $\mathcal{M}$ at $x = g(z)$, denoted as $T_x \mathcal{M}$ (for all $z \in Z$). As GANs are capable of generating realistic samples (particularly for natural images), one can argue that $\mathcal{M}$ approximates the true data manifold well and hence the tangents to $\mathcal{M}$ obtained using $J_z g$ are *close* to the tangents to the true data manifold. The problem of learning a smooth manifold from finite samples has been studied in the literature[5, 2, 27, 6, 40, 19, 14, 3], and it is an interesting problem in its own right to study the manifold approximation error of GANs, which minimize a chosen divergence measure between the data distribution and the *fake* distribution [28, 23] using finite samples, however this is outside the scope of the current work.

For a given data sample $x \in X$, we need to find its corresponding latent representation $z$ before we can use $J_z g$ to get the tangents to the manifold $\mathcal{M}$ at $x$. For our current discussion we assume the availability of a so-called *encoder* $h : X \to Z$, such that $h(g(z)) = z \, \forall \, z \in Z$. By definition, the

Jacobian of the generator at $z$, $J_z g$, can be used to get the tangent directions to the manifold at a point $x = g(z) \in \mathcal{M}$. The following lemma specifies the conditions for existence of the encoder $h$ and shows that such an encoder can also be used to get tangent directions. Later we will come back to the issues involved in training such an encoder.

**Lemma 2.1.** *If the Jacobian of $g$ at $z \in Z$, $J_z g$, is full rank then $g$ is locally invertible in the open neighborhood $g(S)$ ($S$ being an open neighborhood of $z$), and there exists a smooth $h : g(S) \to S$ such that $h(g(y)) = y, \forall y \in S$. In this case, the Jacobian of $h$ at $x = g(z)$, $J_x h$, spans the tangent space of $\mathcal{M}$ at $x$.*

*Proof.* We refer the reader to standard textbooks on multivariate calculus and differentiable manifolds for the first statement of the lemma (e.g., [37]).
The second statement can be easily deduced by looking at the Jacobian of the composition of functions $h \circ g$. We have $J_z(h \circ g) = J_{g(z)} h \, J_z g = J_x h \, J_z g = I_{d \times d}$, since $h(g(z)) = z$. This implies that the row span of $J_x h$ coincides with the column span of $J_z g$. As the columns of $J_z g$ span the tangent space $T_{g(z)} \mathcal{M}$, so do the the rows of $J_x h$. ☐

### 2.1.1 Training the inverse mapping (the encoder)

To estimate the tangents for a given real data point $x \in X$, we need its corresponding latent representation $z = h(x) \in Z$, such that $g(h(x)) = x$ in an ideal scenario. However, in practice $g$ will only learn an approximation to the true data manifold, and the mapping $g \circ h$ will act like a projection of $x$ (which will almost always be off the manifold $\mathcal{M}$) to the manifold $\mathcal{M}$, yielding some approximation error. This projection may not be orthogonal, i.e., to the nearest point on $\mathcal{M}$. Nevertheless, it is desirable that $x$ and $g(h(x))$ are semantically close, and at the very least, the class label is preserved by the mapping $g \circ h$. We studied the following three approaches for training the inverse map $h$, with regard to this desideratum :

- **Decoupled training.** This is similar to an approach outlined by Donahue et al. [8] where the generator is trained first and fixed thereafter, and the encoder is trained by optimizing a suitable reconstruction loss in the $Z$ space, $L(z, h(g(z)))$ (e.g., cross entropy, $\ell_2$). This approach does not yield good results and we observe that most of the time $g(h(x))$ is not semantically similar to the given real sample $x$ with change in the class label. One of the reasons as noted by Donahue et al. [8] is that the encoder never sees real samples during training. To address this, we also experimented with the combined objective $\min_h L_z(z, h(g(z))) + L_h(x, g(h(x)))$, however this too did not yield any significant improvements in our early explorations.
- **BiGAN.** Donahue et al. [8] propose to jointly train the encoder and generator using adversarial training, where the pair $(z, g(z))$ is considered a *fake* example ($z \sim p_z$) and the pair $(h(x), x)$ is considered a *real* example by the discriminator. A similar approach is proposed by Dumoulin et al. [9], where $h(x)$ gives the parameters of the posterior $p(z|x)$ and a stochastic sample from the posterior paired with $x$ is taken as a real example. We use BiGAN [8] in this work, with one modification: we use *feature matching* loss [34] (computed using features from an intermediate layer $\ell$ of the discriminator $f$), i.e., $\|\mathbb{E}_x f_\ell(h(x), x) - \mathbb{E}_z f_\ell(z, g(z))\|_2^2$, to optimize the generator and encoder, which we found to greatly help with the convergence [3]. We observe better results in terms of semantic match between $x$ and $g(h(x))$ than in the decoupled training approach, however, we still observe a considerable fraction of instances where the class of $g(h(x))$ is changed (let us refer to this as *class-switching*).
- **Augmented-BiGAN.** To address the still-persistent problem of *class-switching* of the reconstructed samples $g(h(x))$, we propose to construct a third pair $(h(x), g(h(x))$ which is also considered by the discriminator as a *fake* example in addition to $(z, g(z))$. Our Augmented-BiGAN objective is given as

$$\mathbb{E}_{x \sim p_x} \log f(h(x), x) + \frac{1}{2}\mathbb{E}_{z \sim p_z} \log(1 - f(z, g(z))) + \frac{1}{2}\mathbb{E}_{x \sim p_x} \log(1 - f(h(x), g(h(x)))), \quad (1)$$

where $f(\cdot, \cdot)$ is the probability of the pair being a real example, as assigned by the discriminator $f$. We optimize the discriminator using the above objective (1). The generator and encoder are again optimized using *feature matching* [34] loss on an intermediate layer $\ell$ of the discriminator, i.e., $L_{gh} = \|\mathbb{E}_x f_\ell(h(x), x) - \mathbb{E}_z f_\ell(z, g(z))\|_2^2$, to help with the convergence. Minimizing $L_{gh}$

will make $x$ and $g(h(x))$ similar (through the lens of $f_\ell$) as in the case of BiGAN, however the discriminator tries to make the features at layer $f_\ell$ more difficult to achieve this by directly optimizing the third term in the objective (1). This results in improved semantic similarity between $x$ and $g(h(x))$.

We empirically evaluate these approaches with regard to similarity between $x$ and $g(h(x))$ both quantitatively and qualitatively, observing that Augmented-BiGAN works significantly better than BiGAN. We note that ALI [9] also has the problems of semantic mismatch and *class switching* for reconstructed samples as reported by the authors, and a stochastic version of the proposed third term in the objective (1) can potentially help there as well, investigation of which is left for future work.

### 2.1.2 Estimating the dominant tangent space

Once we have a trained encoder $h$ such that $g(h(x))$ is a good approximation to $x$ and $h(g(z))$ is a good approximation to $z$, we can use either $J_{h(x)}g$ or $J_x h$ to get an estimate of the tangent space. Specifically, the columns of $J_{h(x)}g$ and the rows of $J_x h$ are the directions that approximately span the tangent space to the data manifold at $x$. Almost all deep learning packages implement reverse mode differentiation (to do backpropagation) which is computationally cheaper than forward mode differentiation for computing the Jacobian when the output dimension of the function is low (and vice versa when the output dimension is high). Hence we use $J_x h$ in all our experiments to get the tangents.

As there are approximation errors at several places ($\mathcal{M} \sim$ data-manifold, $g(h(x)) \sim x$, $h(g(z)) \sim z$), it is preferable to only consider dominant tangent directions in the row span of $J_x h$. These can be obtained using the SVD on the matrix $J_x h$ and taking the right singular vectors corresponding to top singular values, as done in [33] where $h$ is trained using a contractive auto-encoder. However, this process is expensive as the SVD needs to be done independently for every data sample. We adopt an alternative approach to get dominant tangent direction: we take the pre-trained model with encoder-generator-discriminator ($h$-$g$-$f$) triple and insert two extra functions $p : R^d \rightarrow R^{d_p}$ and $\bar{p} : R^{d_p} \rightarrow R^d$ (with $d_p < d$) which are learned by optimizing $\min_{p,\bar{p}} \mathbb{E}_x[\|g(h(x)) - g(\bar{p}(p(h(x))))\|_1 + \|f^X_{-1}(g(h(x))) - f^X_{-1}(g(\bar{p}(p(h(x)))))\|]$ while $g$, $h$ and $f$ are kept fixed from the pre-trained model. Note that our discriminator $f$ has two pipelines $f^Z$ and $f^X$ for the latent $z \in Z$ and the data $x \in X$, respectively, which share parameters in the last few layers (following [8]), and we use the last layer of $f^X$ in this loss. This enables us to learn a nonlinear (low-dimensional) approximation in the $Z$ space such that $g(\bar{p}(p(h(x))))$ is close to $g(h(x))$. We use the Jacobian of $p \circ h$, $J_x\, p \circ h$, as an estimate of the $d_p$ dominant tangent directions ($d_p = 10$ in all our experiments)[4].

### 2.2 Injecting invariances into the classifier using tangents

We use the *tangent propagation* approach (TangentProp) [36] to make the classifier invariant to the estimated tangent directions from the previous section. Apart form the regular classification loss on labeled examples, it uses a regularizer of the form $\sum_{i=1}^n \sum_{v \in T_{x_i}} \|(J_{x_i} c)\, v\|_2^2$, where $J_{x_i} c \in \mathbb{R}^{k \times D}$ is the Jacobian of the classifier function $c$ at $x = x_i$ (with the number of classes $k$). and $T_x$ is the set of tangent directions we want the classifier to be invariant to. This term penalizes the *linearized* variations of the classifier output along the tangent directions. Simard et al. [36] get the tangent directions using slight rotations and translations of the images, whereas we use the GAN to estimate the tangents to the data manifold.

We can go one step further and make the classifier invariant to small perturbations in *all* directions emanating from a point $x$. This leads to the regularizer

$$\sup_{v:\|v\|_p \leq \epsilon} \|(J_x c)\, v\|_j^j \leq \sum_{i=1}^k \sup_{v:\|v\|_p \leq \epsilon} |(J_x c)_{i:}\, v|^j = \epsilon^j \sum_{i=1}^k \|(J_x c)_{i:}\|_q^j, \tag{2}$$

where $\|\cdot\|_q$ is the dual norm of $\|\cdot\|_p$ (i.e., $\frac{1}{p} + \frac{1}{q} = 1$), and $\|\cdot\|_j^j$ denotes $j$th power of $\ell_j$-norm. This reduces to squared Frobenius norm of the Jacobian matrix $J_x c$ for $p = j = 2$. The penalty in

Eq. (2) is closely related to the recent work on *virtual adversarial training* (VAT) [22] which uses a regularizer (ref. Eq (1), (2) in [22])

$$\sup_{v:\|v\|_2 \leq \epsilon} \mathrm{KL}[c(x)||c(x+v)], \tag{3}$$

where $c(x)$ are the classifier outputs (class probabilities). VAT[22] approximately estimates $v^*$ that yields the sup using the gradient of $\mathrm{KL}[c(x)||c(x+v)]$, calling $(x+v^*)$ as *virtual adversarial example* (due to its resemblance to *adversarial training* [12]), and uses $\mathrm{KL}[c(x)||c(x+v^*)]$ as the regularizer in the classifier objective. If we replace KL-divergence in Eq. 3 with total-variation distance and optimize its first-order approximation, it becomes equivalent to the regularizer in Eq. (2) for $j = 1$ and $p = 2$.

In practice, it is computationally expensive to optimize these Jacobian based regularizers. Hence in all our experiments we use stochastic finite difference approximation for all Jacobian based regularizers. For TangentProp, we use $\|c(x_i + v) - c(x_i)\|_2^2$ with $v$ randomly sampled (i.i.d.) from the set of tangents $T_{x_i}$ every time example $x_i$ is visited by the SGD. For Jacobian-norm regularizer of Eq. (2), we use $\|c(x + \delta) - c(x)\|_2^2$ with $\delta \sim N(0, \sigma^2 I)$ (i.i.d) every time an example $x$ is visited by the SGD, which approximates an upper bound on Eq. (2) in expectation (up to scaling) for $j = 2$ and $p = 2$.

## 2.3 GAN discriminator as the classifier for semi-supervised learning: effect of fake examples

Recent works have used GANs for semi-supervised learning where the discriminator also serves as a classifier [34, 9, 29]. For a semi-supervised learning problem with $k$ classes, the discriminator has $k + 1$ outputs with the $(k + 1)$'th output corresponding to the *fake* examples originating from the generator of the GAN. The loss for the discriminator $f$ is given as [34]

$$L^f = L^f_{\text{sup}} + L^f_{\text{unsup}}, \text{ where } L^f_{\text{sup}} = -\mathbb{E}_{(x,y) \sim p_d(x,y)} \log p_f(y|x, y \leq k)$$
$$\text{and } L^f_{\text{unsup}} = -\mathbb{E}_{x \sim p_g(x)} \log(p_f(y = k+1|x)) - \mathbb{E}_{x \sim p_d(x)} \log(1 - p_f(y = k+1|x))). \tag{4}$$

The term $p_f(y = k+1|x)$ is the probability of $x$ being a fake example and $(1 - p_f(y = k+1|x))$ is the probability of $x$ being a real example (as assigned by the model). The loss component $L^f_{\text{unsup}}$ is same as the regular GAN discriminator loss with the only modification that probabilities for real vs. fake are compiled from $(k + 1)$ outputs. Salimans et al. [34] proposed training the generator using *feature matching* where the generator minimizes the mean discrepancy between the features for real and fake examples obtained from an intermediate layer $\ell$ of the discriminator $f$, i.e., $L^g = \|\mathbb{E}_x f_\ell(x) - \mathbb{E}_z f_\ell(g(z))\|_2^2$. Using feature matching loss for the generator was empirically shown to result in much better accuracy for semi-supervised learning compared to other training methods including *minibatch discrimination* and regular GAN generator loss [34].

Here we attempt to develop an intuitive understanding of how fake examples influence the learning of the classifier and why feature matching loss may work much better for semi-supervised learning compared to regular GAN. We will use the term classifier and discriminator interchangeably based on the context however they are really the same network as mentioned earlier. Following [34] we assume the $(k + 1)$'th logit is fixed to $0$ as subtracting a term $v(x)$ from all logits does not change the *softmax* probabilities. Rewriting the unlabeled loss of Eq. (4) in terms of *logits* $l_i(x)$, $i = 1, 2, \ldots, k$, we have

$$L^f_{\text{unsup}} = \mathbb{E}_{x_g \sim p_g} \log\left(1 + \sum_{i=1}^{k} e^{l_i(x_g)}\right) - \mathbb{E}_{x \sim p_d}\left[\log \sum_{i=1}^{k} e^{l_i(x)} - \log\left(1 + \sum_{i=1}^{k} e^{l_i(x)}\right)\right] \tag{5}$$

Taking the derivative w.r.t. discriminator's parameters $\theta$ followed by some basic algebra, we get $\nabla_\theta L^f_{\text{unsup}} =$

$$\mathbb{E}_{x_g \sim p_g} \sum_{i=1}^{k} p_f(y = i|x_g) \nabla l_i(x_g) - \mathbb{E}_{x \sim p_d}\left[\sum_{i=1}^{k} p_f(y = i|x, y \leq k)\nabla l_i(x) - \sum_{i=1}^{k} p_f(y = i|x)\nabla l_i(x)\right]$$

$$= \mathbb{E}_{x_g \sim p_g} \sum_{i=1}^{k} \underbrace{p_f(y = i|x_g)}_{a_i(x_g)} \nabla l_i(x_g) - \mathbb{E}_{x \sim p_d} \sum_{i=1}^{k} \underbrace{p_f(y = i|x, y \leq k)p_f(y = k+1|x)}_{b_i(x)} \nabla l_i(x) \tag{6}$$

Minimizing $L_{\text{unsup}}^f$ will move the parameters $\theta$ so as to decrease $l_i(x_g)$ and increase $l_i(x)$ ($i = 1, \ldots, k$). The rate of increase in $l_i(x)$ is also modulated by $p_f(y = k+1|x)$. This results in warping of the functions $l_i(x)$ around each real example $x$ with more warping around examples about which the current model $f$ is more confident that they belong to class $i$: $l_i(\cdot)$ becomes locally concave around those real examples $x$ if $x_g$ are loosely scattered around $x$. Let us consider the following three cases:

**Weak fake examples.** When the fake examples coming from the generator are very *weak* (i.e., very easy for the current discriminator to distinguish from real examples), we will have $p_f(y = k+1|x_g) \approx 1$, $p_f(y = i|x_g) \approx 0$ for $1 \le i \le k$ and $p_f(y = k+1|x) \approx 0$. Hence there is no gradient flow from Eq. (6), rendering unlabeled data almost useless for semi-supervised learning.

**Strong fake examples.** When the fake examples are very *strong* (i.e., difficult for the current discriminator to distinguish from real ones), we have $p_f(k+1|x_g) \approx 0.5 + \epsilon_1$, $p_f(y = i_{\max}|x_g) \approx 0.5 - \epsilon_2$ for some $i_{\max} \in \{1, \ldots, k\}$ and $p_f(y = k+1|x) \approx 0.5 - \epsilon_3$ (with $\epsilon_2 > \epsilon_1 \ge 0$ and $\epsilon_3 \ge 0$). Note that $b_i(x)$ in this case would be smaller than $a_i(x)$ since it is a product of two probabilities. If two examples $x$ and $x_g$ are close to each other with $i_{\max} = \arg\max_i l_i(x) = \arg\max_i l_i(x_g)$ (e.g., $x$ is a *cat* image and $x_g$ is a highly realistic generated image of a *cat*), the optimization will push $l_{i_{\max}}(x)$ up by some amount and will pull $l_{i_{\max}}(x_g)$ down by a larger amount. We further want to consider two cases here: **(i) Classifier with enough capacity:** If the classifier has enough capacity, this will make the curvature of $l_{i_{\max}}(\cdot)$ around $x$ really high (with $l_{i_{\max}}(\cdot)$ locally concave around $x$) since $x$ and $x_g$ are very close. This results in over-fitting around the unlabeled examples and for a test example $x_t$ closer to $x_g$ (which is quite likely to happen since $x_g$ itself was very realistic sample), the model will more likely misclassify $x_t$. **(ii) Controlled-capacity classifier:** Suppose the capacity of the classifier is controlled with adequate regularization. In that case the curvature of the function $l_{i_{\max}}(\cdot)$ around $x$ cannot increase beyond a point. However, this results in $l_{i_{\max}}(x)$ being pulled down by the optimization process since $a_i(x_g) > b_i(x)$. This is more pronounced for examples $x$ on which the classifier is not so confident (i.e., $p_f(y = i_{\max}|x, y \le k)$ is low, although still assigning highest probability to class $i_{\max}$) since the gap between $a_i(x_g)$ and $b_i(x)$ becomes higher. For these examples, the entropy of the distribution $\{p(y = i|x, y \le k)\}_{i=1}^k$ may actually *increase* as the training proceeds which can hurt the test performance.

**Moderate fake examples.** When the fake examples from the generator are neither too weak nor too strong for the current discriminator (i.e., $x_g$ is a somewhat distorted version of $x$), the unsupervised gradient will push $l_{i_{\max}}(x)$ up while pulling $l_{i_{\max}}(x_g)$ down, giving rise to a moderate curvature of $l_i(\cdot)$ around real examples $x$ since $x_g$ and $x$ are sufficiently far apart (consider multiple distorted *cat* images scattered around a real *cat* image at moderate distances). This results in a smooth decision function around real unlabeled examples. Again, the curvatures of $l_i(\cdot)$ around $x$ for classes $i$ which the current classifier does not trust for the example $x$ are not affected much. Further, $p_f(y = k+1|x)$ will be less than the case when fake examples are very strong. Similarly $p_f(y = i_{\max}|x_g)$ (where $i_{\max} = \arg\max_{1 \le i \le k} l_i(x_g)$) will be less than the case of strong fake examples. Hence the norm of the gradient in Eq. (6) is lower and the contribution of unlabeled data in the overall gradient of $L^f$ (Eq. (4) is lower than the case of strong fake examples. This intuitively seems beneficial as the classifier gets ample opportunity to learn on supervised loss and get confident on the right class for unlabeled examples, and then boost this confidence slowly using the gradient of Eq. (6) as the training proceeds.

We experimented with regular GAN loss (i.e., $L^g = \mathbb{E}_{x \sim p_g} \log(p_f(y = k+1|x)))$, and feature matching loss for the generator [34], plotting several of the quantities of interest discussed above for MNIST (with 100 labeled examples) and SVHN (with 1000 labeled examples) datasets in Fig.1. Generator trained with feature matching loss corresponds to the case of *moderate fake examples* discussed above (as it generates blurry and distorted samples as mentioned in [34]). Generator trained with regular GAN loss corresponds to the case of *strong fake examples* discussed above. We plot $\mathbb{E}_{x_g} a_{i_{\max}}(x_g)$ for $i_{\max} = \arg\max_{1 \le i \le k} l_i(x_g)$ and $\mathbb{E}_{x_g}[\frac{1}{k-1} \sum_{1 \le i \ne i_{\max} \le k} a_i(x_g)]$ separately to look into the behavior of $i_{\max}$ logit. Similarly we plot $\mathbb{E}_x b_t(x)$ separately where $t$ is the true label for unlabeled example $x$ (we assume knowledge of the true label only for plotting these quantities and not while training the semi-supervised GAN). Other quantities in the plots are self-explanatory. As expected, the unlabeled loss $L_{\text{unsup}}^f$ for regular GAN becomes quite high early on implying that fake examples are strong. The gap between $a_{i_{\max}}(x_g)$ and $b_t(x)$ is also higher for regular GAN pointing towards the case of *strong fake examples with controlled-capacity classifier* as discussed above. Indeed, we see that the average of the entropies for the distributions $p_f(y|x)$ (i.e.,

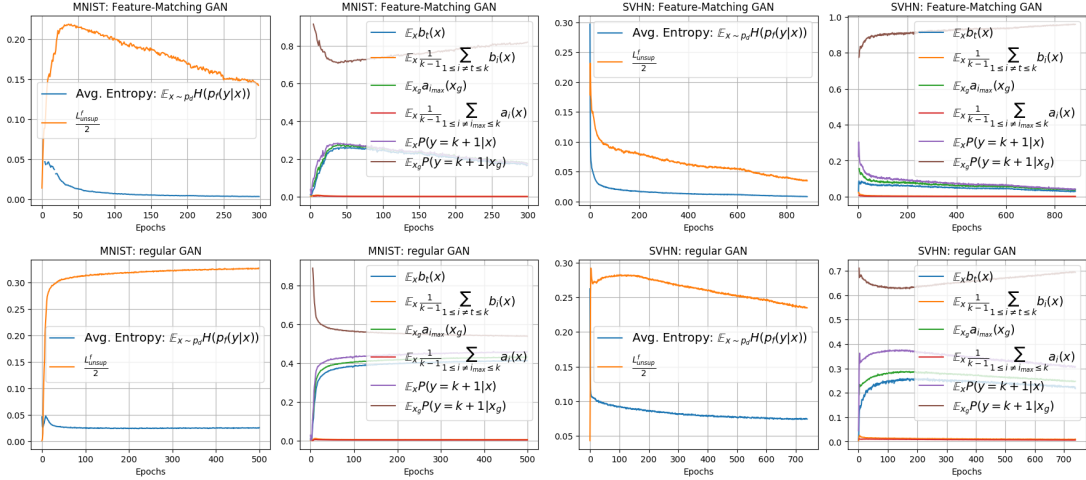

Figure 1: Plots of Entropy, $L_{\text{unsup}}^f$ (Eq. (4)), $a_i(x_g)$, $b_i(x)$ and other probabilities (Eq. (6)) for regular GAN generator loss and feature-matching GAN generator loss.

$\mathbb{E}_x H(p_f(y|x, y \leq k)))$ is much lower for feature-matching GAN compared to regular GAN (seven times lower for SVHN, ten times lower for MNIST). Test errors for MNIST for regular GAN and FM-GAN were $2.49\%$ (500 epochs) and $0.86\%$ (300 epochs), respectively. Test errors for SVHN were $13.36\%$ (regular-GAN at 738 epochs) and $5.89\%$ (FM-GAN at 883 epochs), respectively[5]. It should also be emphasized that the semi-supervised learning heavily depends on the generator dynamically adapting fake examples to the current discriminator – we observed that freezing the training of the generator at any point results in the discriminator being able to classify them easily (i.e., $p_f(y = k + 1|x_g) \approx 1$) thus stopping the contribution of unlabeled examples in the learning.

**Our final loss for semi-supervised learning.** We use feature matching GAN with semi-supervised loss of Eq. (4) as our classifier objective and incorporate invariances from Sec. 2.2 in it. Our final objective for the GAN discriminator is

$$L^f = L_{\text{sup}}^f + L_{\text{unsup}}^f + \lambda_1 \mathbb{E}_{x \sim p_d(x)} \sum_{v \in T_x} \|(J_x f) v\|_2^2 + \lambda_2 \mathbb{E}_{x \sim p_d(x)} \|J_x f\|_F^2. \tag{7}$$

The third term in the objective makes the classifier decision function change slowly along tangent directions around a real example $x$. As mentioned in Sec. 2.2 we use stochastic finite difference approximation for both Jacobian terms due to computational reasons.

## 3 Experiments

**Implementation Details.** The architecture of the endoder, generator and discriminator closely follow the network structures in ALI [9]. We remove the stochastic layer from the ALI encoder (i.e., $h(x)$ is deterministic). For estimating the dominant tangents, we employ fully connected two-layer network with $tanh$ non-linearly in the hidden layer to represent $p \circ \bar{p}$. The output of $p$ is taken from the hidden layer. Batch normalization was replaced by weight normalization in all the modules to make the output $h(x)$ (similarly $g(z)$) dependent only on the given input $x$ (similarly $z$) and not on the whole minibatch. This is necessary to make the Jacobians $J_x h$ and $J_z g$ independent of other examples in the minibatch. We replaced all ReLU nonlinearities in the encoder and the generator with the Exponential Linear Units (ELU) [7] to ensure smoothness of the functions $g$ and $h$. We follow [34] completely for optimization (using ADAM optimizer [15] with the same learning rates as in [34]). Generators (and encoders, if applicable) in all the models are trained using feature matching loss.

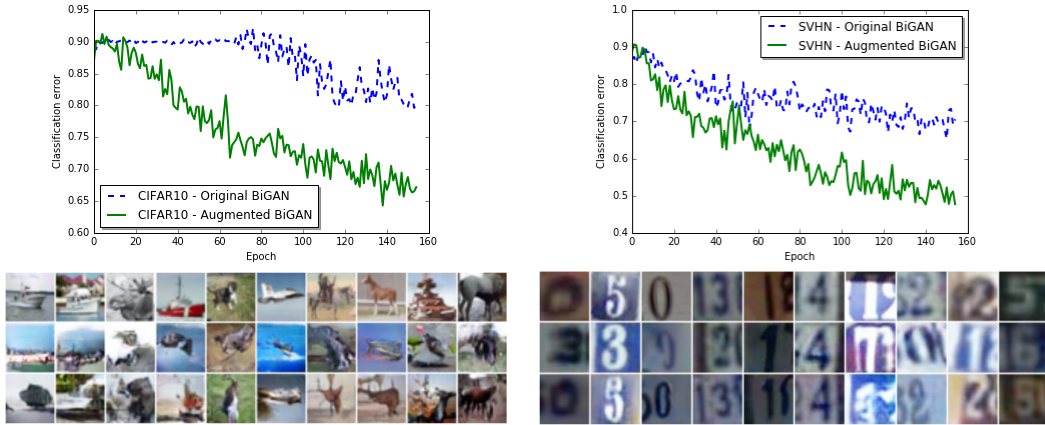

Figure 2: Comparing BiGAN with Augmented BiGAN based on the classification error on the reconstructed test images. *Left column:* CIFAR10, *Right column:* SVHN. In the images, the top row corresponds to the original images followed by BiGAN reconstructions in the middle row and the Augmented BiGAN reconstructions in the bottom row. More images can be found in the appendix.

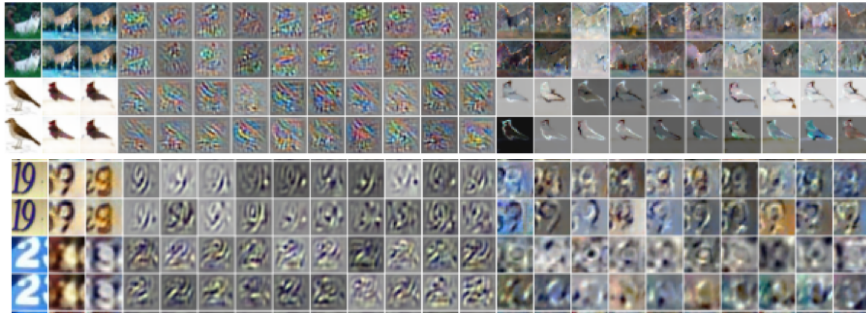

Figure 3: Visualizing tangents. *Top:* CIFAR10, *Bottom: SVHN. Odd rows:* Tangents using our method for estimating the dominant tangent space. *Even rows:* Tangents using SVD on $J_{h(x)}g$ and $J_xh$. *First column:* Original image. *Second column:* Reconstructed image using $g \circ h$. *Third column:* Reconstructed image using $g \circ \bar{p} \circ p \circ h$. *Columns 4-13:* Tangents using encoder. *Columns 14-23:* Tangents using generator.

**Semantic Similarity.** The image samples $x$ and their reconstructions $g(h(x))$ for BiGAN and Augemented-BiGAN can be seen in Fig. 2. To quantitatively measure the semantic similarity of the reconstructions to the original images, we learn a supervised classifier using the full training set and obtain the classification accuracy on the reconstructions of the test images. The architectures of the classifier for CIFAR10 and SVHN are similar to their corresponding GAN discriminator architectures we have. The lower error rates with our Augmented-BiGAN suggest that it leads to reconstructions with reduced *class-switching*.

**Tangent approximations.** Tangents for CIFAR10 and SVHN are shown in Fig. 3. We show visual comparison of tangents from $J_x(p \circ h)$, from $J_{p(h(x))}g \circ \bar{p}$, and from $J_xh$ and $J_{h(x)}g$ followed by the SVD to get the dominant tangents. It can be seen that the proposed method for getting dominant tangent directions gives similar tangents as SVD. The tangents from the generator (columns 14-23) look different (more colorful) from the tangents from the encoder (columns 4-13) though they do trace the boundaries of the objects in the image (just like the tangents from the encoder). We also empirically quantify our method for dominant tangent subspace estimation against the SVD estimation by computing the geodesic distances and principal angles between these two estimations. These results are shown in Table 2.

**Semi-supervised learning results.** Table 1 shows the results for SVHN and CIFAR10 with various number of labeled examples. For all experiments with the tangent regularizer for both CIFAR10 and SVHN, we use 10 tangents. The hyperparameters $\lambda_1$ and $\lambda_2$ in Eq. (7) are set to 1. We obtain significant improvements over baselines, particularly for SVHN and more so for the case of 500

| Model | SVHN | | CIFAR-10 | |
|---|---|---|---|---|
| | $N_l = 500$ | $N_l = 1000$ | $N_l = 1000$ | $N_l = 4000$ |
| VAE (M1+M2) [17] | – | $36.02 \pm 0.10$ | – | – |
| SWWAE with dropout [41] | – | $23.56$ | – | – |
| VAT [22] | – | $24.63$ | – | – |
| Skip DGM [20] | – | $16.61 \pm 0.24$ | – | – |
| Ladder network [32] | – | – | – | $20.40$ |
| ALI [9] | – | $7.41 \pm 0.65$ | $19.98 \pm 0.89$ | $17.99 \pm 1.62$ |
| FM-GAN [34] | $18.44 \pm 4.8$ | $8.11 \pm 1.3$ | $21.83 \pm 2.01$ | $18.63 \pm 2.32$ |
| Temporal ensembling [18] | $5.12 \pm 0.13$ | $4.42 \pm 0.16$ | – | $\mathbf{12.16} \pm 0.24$ |
| FM-GAN + Jacob.-reg (Eq. (2)) | $10.28 \pm 1.8$ | $4.74 \pm 1.2$ | $20.87 \pm 1.7$ | $16.84 \pm 1.5$ |
| FM-GAN + Tangents | $5.88 \pm 1.5$ | $5.26 \pm 1.1$ | $20.23 \pm 1.3$ | $16.96 \pm 1.4$ |
| FM-GAN + Jacob.-reg + Tangents | $\mathbf{4.87} \pm 1.6$ | $\mathbf{4.39} \pm 1.2$ | $\mathbf{19.52} \pm 1.5$ | $16.20 \pm 1.6$ |

Table 1: Test error with semi-supervised learning on SVHN and CIFAR-10 ($N_l$ is the number of labeled examples). All results for the proposed methods (last 3 rows) are obtained with training the model for 600 epochs for SVHN and 900 epochs for CIFAR10, and are averaged over 5 runs.

| | $d(\mathcal{S}_1, \mathcal{S}_2)$ | $\theta_1$ | $\theta_2$ | $\theta_3$ | $\theta_4$ | $\theta_5$ | $\theta_6$ | $\theta_7$ | $\theta_8$ | $\theta_9$ | $\theta_{10}$ |
|---|---|---|---|---|---|---|---|---|---|---|---|
| Rand-Rand | 4.5 | 14 | 83 | 85 | 86 | 87 | 87 | 88 | 88 | 88 | 89 |
| SVD-Approx. (CIFAR) | 2.6 | 2 | 15 | 21 | 26 | 34 | 40 | 50 | 61 | 73 | 85 |
| SVD-Approx. (SVHN) | 2.3 | 1 | 7 | 12 | 16 | 22 | 30 | 41 | 51 | 67 | 82 |

Table 2: Dominant tangent subspace approximation quality: Columns show the geodesic distance and 10 principal angles between the two subspaces. *Top row* shows results for two randomly sampled 10-dimensional subspaces in 3072-dimensional space, *middle* and *bottom* rows show results for dominant subspace obtained using SVD of $J_x h$ and dominant subspace obtained using our method, for CIFAR-10 and SVHN, respectively. All numbers are averages 10 randomly sampled test examples.

labeled examples. We do not get as good results on CIFAR10 which may be due to the fact that our encoder for CIFAR10 is still not able to approximate the inverse of the generator well (which is evident from the sub-optimal reconstructions we get for CIFAR10) and hence the tangents we get are not good enough. We think that obtaining better estimates of tangents for CIFAR10 has the potential for further improving the results. ALI [9] accuracy for CIFAR ($N_l = 1000$) is also close to our results however ALI results were obtained by running the optimization for 6475 epochs with a slower learning rate as mentioned in [9]. Temporal ensembling [18] using explicit data augmentation assuming knowledge of the class-preserving transformations on the input, while our method estimates these transformations from the data manifold in the form of tangent vectors. It outperforms our method by a significant margin on CIFAR-10 which could be due the fact that it uses horizontal flipping based augmentation for CIFAR-10 which cannot be learned through the tangents as it is a non-smooth transformation. The use of temporal ensembling in conjunction with our method has the potential of further improving the semi-supervised learning results.

## 4  Discussion

Our empirical results show that using the tangents of the data manifold (as estimated by the generator of the GAN) to inject invariances in the classifier improves the performance on semi-supevised learning tasks. In particular we observe impressive accuracy gains on SVHN (more so for the case of 500 labeled examples) for which the tangents obtained are good quality. We also observe improvements on CIFAR10 though not as impressive as SVHN. We think that improving on the quality of tangents for CIFAR10 has the potential for further improving the results there, which is a direction for future explorations. We also shed light on the effect of fake examples in the common framework used for semi-supervised learning with GANs where the discriminator predicts *real* class labels along with the *fake* label. Explicitly controlling the difficulty level of fake examples (i.e., $p_f(y = k + 1|x_g)$ and hence indirectly $p_f(y = k + 1|x)$ in Eq. (6)) to do more effective semi-supervised learning is another direction for future work. One possible way to do this is to have a distortion model for the real examples (i.e., replace the *generator* with a *distorter* that takes as input the real examples) whose strength is controlled for more effective semi-supervised learning.

## Footnotes

[2] We write $g$ as a map from $Z$ to $X$ to avoid the unnecessary (in our context) burden of manifold terminologies and still being technically correct. This also enables us to get the Jacobian of $g$ as a regular matrix in $\mathbb{R}^{D \times d}$, instead of working with the *differential* if $g$ was taken as a map from $Z$ to $\mathcal{M}$.

[3]Note that other recently proposed methods for training GANs based on Integral Probability Metrics [1, 13, 26, 24] could also improve the convergence and stability during training.

[4]Training the GAN with $z \in Z \subset \mathbb{R}^{d_p}$ results in a bad approximation of the data manifold. Hence we first learn the GAN with $Z \subset \mathbb{R}^d$ and then approximate the smooth manifold $\mathcal{M}$ parameterized by the generator using $p$ and $\bar{p}$ to get the dominant $d_p$ tangent directions to $\mathcal{M}$.

[5]We also experimented with minibatch-discrimination (MD) GAN[34] but the minibatch features are not suited for classification as the prediction for an example $x$ is adversely affected by features of all other examples (note that this is different from batch-normalization). Indeed we notice that the training error for MD-GAN is 10x that of regular GAN and FM-GAN. MD-GAN gave similar test error as regular-GAN.

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
