[Supplementary Material]

# Appendix

## A   Tangent Plots

Figure 4: CIFAR10 tangents. *Odd rows:* Tangents using our method for estimating the dominant tangent space. *Even rows:* Tangents using SVD on $J_{h(x)}g$ and $J_x h$. *First column:* Original image. *Second column:* Reconstructed image using $g \circ h$. *Third column:* Reconstructed image using $g \circ \bar{p} \circ p \circ h$. *Columns 4-13:* Tangents using encoder. *Columns 14-23:* Tangents using generator.

Figure 5: SVHN tangents. *Odd rows:* Tangents using our method for estimating the dominant tangent space. *Even rows:* Tangents using SVD on $J_{h(x)}g$ and $J_x h$. *First column:* Original image. *Second column:* Reconstructed image using $g \circ h$. *Third column:* Reconstructed image using $g \circ \bar{p} \circ p \circ h$. *Columns 4-13:* Tangents using encoder. *Columns 14-23:* Tangents using generator.

# B  Reconstruction Plots

Figure 6: CIFAR10 reconstructions: Comparing BiGAN reconstructions with Augmented-BiGAN reconstructions. For $i = 0, 1, \ldots, 4$, $(3i + 1)$th row shows the original images, followed by BiGAN reconstructions in the $(3i + 2)$'th row, and the Augmented-BiGAN reconstructions in the $(3i + 3)$'th row. Reconstructions shown for total 100 randomly sampled images from the test set ($10 \times 5 = 50$ images each in the left and right column) from the test set.

Figure 7: SVHN reconstructions: Comparing BiGAN reconstructions with Augmented-BiGAN reconstructions. For $i = 0, 1, \ldots, 4$, $(3i + 1)$th row shows the original images, followed by BiGAN reconstructions in the $(3i + 2)$'th row, and the Augmented-BiGAN reconstructions in the $(3i + 3)$'th row. Reconstructions shown for total 100 randomly sampled images from the test set ($10 \times 5 = 50$ images each in the left and right column).