[Reviews · NeurIPS 2017]

Reviewer 1



The author(s) extend the idea of regularizing classifiers to be invariant to the tangent space of the learned manifold of the data to use GAN based architectures. This is a worthwhile idea to revisit as significant advances have been made in generative modeling in the intervening time since the last major paper in the area, the CAE was published. Crucial to the idea is the existence of an encoder learning an inverse mapping of the standard generator of GAN training. This is still an area of active research in the GAN literature that as of yet has no completely satisfactory approach. As current inference techniques for GANs are still quite poor, the authors propose two improvements to one technique, BiGAN, which are worthwhile contributions. 1) They adopt the feature matching loss proposed in "Improved techniques for training gans" and 2) they augment the BiGAN objective with another term that evaluates how the generator maps the inferred latent code for a given real example. Figure 2 provides good evidence of the usefulness of these modifications. However, even with the demonstrated improvements, the label switching problem is still very significant. The reconstructed CIFAR10 examples using their approach are only classified correctly ~35% of the time which causes me to wonder whether the approximation errors of current approaches are still too high to realize the potential benefits of the approach. In addition, the visualized tangents are qualitatively quite different for the encoder and generator which is also suggestive of significant approximation error. Nonetheless, the empirical results suggests benefits can be realized even with current approximation issues. The results are much stronger for the case of the simpler data manifold of SVHN (improving the FM-GAN baseline from 18.44% to 6.6% error in the case of 500 labeled examples) compared to CIFAR-10 where consistent, but much smaller benefits, are seen. Could the author(s) speculate or comment on why even with all the approximations of the current approach why their system still performs reasonably? The paper appears to have struggled with the 8 page manuscript limit and abruptly cuts off after the empirical results with no concluding section.

Reviewer 2



Chiefly theoretical work with some empirical results on SVHN and CIFAR10. This paper proposes using a trained GAN to estimate mappings from and to the true data distribution around a data point, and use a kind of neural PCA to estimate tangents to those estimates, then used for training a manifold-invariant classifier. Some additional work investigating regular GAN vs feature matching GAN is briefly presented, and an augmentation to BiGAN. It feels a bit like this is two works squeezed into one. Maybe three. There is an exploration of FM-GAN vs a regular GAN training objective, with some nice results shown where the classification entropy (confidence of the classification, as I read it) is much better for an FM-GAN than for a regular GAN. There is an Augmented BiGAN which achieves nicely lower classification error than BiGAN on infer-latents-then-generate g(h(x)). The most substantial work presented here is the manifold-invariance. The first thing I wrestle with is that the method is a bit complex, making it probably tricky for others to get right, and complex/fiddly to implement. In particular, 2.1.2 proposes to freeze f, g, and h, and introduce p and pbar as a nonlinear approximation to SVD. This introduces the second thing I wrestle with: numerous layered approximations. The method requires g and h to be good approximations to generate and infer to and from the data manifold. The results (e.g. figure 2) do not indicate that these approximations are always very good. The method requires that p and pbar reasonably capture singular directions in the latent space, but figure 3 shows this approximation only sort-of holds. This has me wondering about transferability, and wondering how to measure each of the approximation errors to evaluate whether the method is useful for a dataset. The CIFAR-10 results reinforce my concern. The MNIST result in line 264 (0.86) is quite good for semi-supervised. What is the difference between the results in 264-265 vs the results in Table 1? Different numbers are given for SVHN in each location, yet line 251 suggests the results in both locations are semi-sup. Work would feel more complete with comparisons on semi-sup MNIST in Table 1, then you could show ladder, CAE, MTC, etc. I'm guessing you're up against space constraints here... Table 1 missing some competitive results. A quick Google search for svhn semi supervised gives https://arxiv.org/abs/1610.02242 showing 7.05% for SVHN with 500 labels, 5.43% with 1000 labels; 16.55% CIFAR10@4k. https://papers.nips.cc/paper/6333-regularization-with-stochastic-transformations-and-perturbations-for-deep-semi-supervised-learning.pdf reports 6.03% on SVHN with 1% of labels (~700). Missing conclusion/future directions. Minor nitpicks: grammar of lines 8-9 needs work grammar of lines 42-43 lines 72/73 have two citations in a row lines 88/89 unnecessary line break line 152 wrong R symbol line 204 comma before however lne 295 'do not get as better' grammar needs work All in all, the results are OK-to-good but not universally winning. The topic is of interest to the community, including a few novel ideas. (The paper perhaps should be multiple works.) It looks like a new SOTA is presented for SVHN semisup, which tips me toward accept.

Reviewer 3



This paper describes a method for semi-supervised learning which is both adversarial and promotes the classifier's robustness to input variations. The GAN-based semi-supervised framework adopted in the paper is standard, treating the generated samples as an additional class to the regular classes that the classifier aims to label. What is new is the Jacobian-based regularizations that are introduced to encourage the classifier to be robust to local variations in the tangent space of the input manifold. The paper proposes an efficient method for estimating the tangents space at each training sample, avoiding the expensive SVD-based method used in contractive autoencoders. The paper also proposed an improved version of BiGAN, called Augmented-BiGAN, for training the encoder used in calculating the Jacobians. The technical development culminates in a semi-supervised objective that simultaneously incorporates classification of labeled samples, adversarial generative learning of unlabeled/labeled samples, and variation penalties that encourages smoothness of the classifier in the input manifold. The experiments are based on CIFAR10 and SVHN, showing the benefits of Augmented-BiGAN over the original BiGAN and the performance improvements on semi-supervised learning due to the Jacobian-based regularizations. The discussions on the experiments could be expanded. A detailed comparison to ALI could be added to make the results more comprehensive.